# Genome-wide association study and pathway analysis identify NTRK2 as a novel candidate gene for litter size in sheep

**Seyed Mehdi Esmaeili-Fard**[1]*, **Mohsen Gholizadeh**[1], **Seyed Hasan Hafezian**[1], **Rostam Abdollahi-Arpanahi**[2]

**1** Department of Animal Sciences and Fisheries, Sari Agricultural Sciences and Natural Resources University (SANRU), Sari, Iran, **2** Department of Animal and Dairy Science, University of Georgia, Athens, GA, United States of America

* mehdi.esmaeilifard@gmail.com

**Data Availability Statement:** The datasets supporting the conclusions of the manuscript are available in the Figshare repository under the

## Abstract

Litter size is one of the most important economic traits in sheep. Identification of gene variants that are associated with the prolificacy rate is an important step in breeding program success and profitability of the farm. So, to identify genetic mechanisms underlying the variation in litter size in Iranian Baluchi sheep, a two-step genome-wide association study (GWAS) was performed. GWAS was conducted using genotype data from 91 Baluchi sheep. Estimated breeding values (EBVs) for litter size calculated for 3848 ewes and then used as the response variable. Besides, a pathway analysis using GO and KEGG databases were applied as a complementary approach. A total of three single nucleotide polymorphisms (SNPs) associated with litter size were identified, one each on OAR2, OAR10, and OAR25. The SNP on OAR2 is located within a novel putative candidate gene, Neurotrophic receptor tyrosine kinase 2. This gene product works as a receptor which is essential for follicular assembly, early follicular growth, and oocyte survival. The SNP on OAR25 is located within RAB4A which is involved in blood vessel formation and proliferation through angiogenesis. The SNP on OAR10 was not associated with any gene in the 1Mb span. Moreover, gene-set analysis using the KEGG database identified several pathways, such as Ovarian steroidogenesis, Steroid hormone biosynthesis, Calcium signaling pathway, and Chemokine signaling. Also, pathway analysis using the GO database revealed several functional terms, such as cellular carbohydrate metabolic, biological adhesion, cell adhesion, cell junction, and cell-cell adherens junction, among others. This is the first study that reports the *NTRK2* gene affecting litter size in sheep and our study of this gene functions showed that this gene could be a good candidate for further analysis.

## Introduction

Flock profitability is greatly affected by ewe reproductive efficiency. The outcome of reproductive traits in sheep is quite different from one breed to another and also within breeds.

following DOI number: https://doi.org/10.6084/m9.figshare.12515189.v1

**Funding:** The author(s) received no specific funding for this work.

**Competing interests:** The authors have declared that no competing interests exist.

Identification of ewes with higher prolificacy rates is an important step in sheep breeding success and profitability of the farm [1]. In sheep, understanding the genetic basis for high prolificacy is valuable for genetic improvement and management and provides basic knowledge that may facilitate modification of fertility and prolificacy in the future [2]. The heritability of the reproductive traits varies from low to moderate and, therefore, response to selection under classic phenotypic breeding programs is expected to be low. As an alternative, identifying major gene variants that affect different parts of the reproductive process, such as ovulation rate and litter size could be very helpful and used in marker-assisted selection (MAS) context [3]. Characteristics of a major gene affecting prolificacy in a population include high variation in ovulation rate and litter size, combined with high repeatability. Reproduction is a complex process, and traits such as ovulation rate and litter size are genetically affected by many minor genes and also some major genes, called fecundity (Fec) genes [4]. To date, a number of this fecundity genes have been identified in sheep, including *GDF9* [5], *BMP15* [6], *BMPR1B* [7], *B4GALNT2* [8], *FecX2$^W$* [9], and *Fec$^D$* [10] located on ovine chromosomes 5, X, 6, 11, X, and an unknown autosome, respectively. So far, several studies using different methods have been performed to identify gene variants affecting litter size in Iranian sheep breeds, but nearly none of the effective variants of these known genes have been identified yet [1, 11–13]. However as some of the identified variants might be population-specific, further scanning of the genome is needed to identify additional strong candidate genes in the population of interest.

Baluchi sheep are one of the Iranian fat-tailed breeds and comprise approximately 30% of the country's total sheep inventory. This breed is well-adapted to the arid subtropical environments of eastern Iran and can play an important role in supplying the need for meat due to its higher twinning rate than other Iranian breeds and acceptable growth rates [14, 15]. To date, only one genome-wide association study on the litter size in Baluchi sheep was conducted. Gholizadeh et al. [1] using a GLM model and birth type records as the response variable, performed a whole-genome scan for each parity separately and identified two significant single nucleotide polymorphisms (SNPs), rs407696726 and rs412433416, on OAR10 and OAR15. The SNP on OAR15 is located within the *DYNC2H1* gene, but SNP on OAR10 was not associated with any gene in a 1 Mb span. However, in the present study, we used a two-step GWAS accompanied by a gene-set analysis (GSA) as a complementary approach to identify genes and pathways affecting litter size in Iranian Baluchi sheep. Here, the estimated breeding values (EBVs) were used as the response variable.

Complex traits are controlled by several genes and therefore, association studies identify only the most significant SNPs (due to using stringent threshold) and their neighboring genes that signify only a small portion of the genetic variation [16]. So, several poorly associated SNPs are always ignored and the large majority of the genetic variants contributing to the trait remains hidden [17, 18]. An alternative strategy to tackle the aforesaid problem is to move up the analysis from the SNP to the gene and gene-set levels [19]. gene set or pathway-based analysis has been proposed as a complementary approach to investigate complex traits from a genetic and biological perspective and work based on evaluating modules of functionally related genes, rather than focusing only on the most significant markers [16, 20]. In this approach, a set of related genes, such as genes in a specific pathway, that harbor significant SNPs previously detected in GWAS (although with a less stringent threshold), is tested for over-representation in a specific pathway [21].

Failure to identify known genes affecting litter size in Baluchi breeds as well as other Iranian sheep breeds reinforces the possibility that another mechanism may be involved in the litter size variation in Iranian sheep. So, to unravel the genomic architecture underlying the litter size in Baluchi sheep, a GWA analysis using EBVs as a response variable was performed. Gene-set analysis was used as a complementary approach for GWAS.

## Materials and methods

### Phenotypic and genotypic data

All the required information for this study was provided by the Department of Animal Science of Sari Agriculture Science and Natural Resource University (SANRU), Iran [1]. The phenotype dataset consisted of 3,848 birth records from 1,506 ewes that were collected from 2004 to 2012 at Abbas Abad Baluchi sheep Breeding Station, Iran. Descriptive statistics of the litter size trait is presented in Table 1. The pedigree file encompassed 4,727 animals with 178 sires, 1,509 dams, and 818 founders.

A total of 91 of 1,506 ewes were genotyped by Illumina 50K SNP panel for 54,241 markers. A similar number of single- and twin-bearing ewes with minimum pedigree relationships to one another were selected for genotyping. These ewes had a total of 435 repeated records across six years.

Quality control of SNP genotypes was performed in GenABEL [22] R [23] package. The SNPs which had minor allele frequency (MAF) less than 0.01, genotype call rate less than 93%, Hardy–Weinberg equilibrium deviations less than $10^{-6}$ p-value cutoff, and SNPs without known genomic location were removed. Also, ewes with a genotyping call rate of less than 95%, were excluded from the dataset.

### Genome-wide association study

For GWA analysis, a two-step approach was used. In the first step, the estimated breeding values of all ewes for the litter size trait were calculated using the repeatability BLUP (Best linear unbiased prediction) model. To do this, a primary GLM model was run in R software to identify significant fixed effects on litter size. Then the following model was used for EBV calculation:

$$y = Xb + Zu + Wpe + e$$

where, $y$ is a vector of birth type (1 and 2); $b$ is a vector of fixed effects including dam age (2–6 years), birth year (2006–2012), and parities (1–5); $u$ is the vector of random additive direct genetic effects, $pe$ is the vector of permanent environmental effects, and $e$ is the vector of random residual effects. The matrices $\mathbf{X}$, $\mathbf{Z}$, and $\mathbf{W}$ are the incidence matrices relating animal records to fixed and random effects, respectively. In this model, the random effects have multivariate Gaussian (co)variance,

$$\left( \begin{matrix} u \\ pe \\ e \end{matrix} \Big| \sigma_u^2, \sigma_{pe}^2, \sigma_e^2 \right) \sim N \left[ 0, \begin{pmatrix} A\sigma_u^2 & 0 & 0 \\ 0 & I_n\sigma_{pe}^2 & 0 \\ 0 & 0 & I_N\sigma_e^2 \end{pmatrix} \right]$$

Where $A$ is the pedigree-based relationship matrix; $I$ is the identity matrix; $n$ is the number of ewes in the dataset (n = 1,506) and $N$ is the total number of records (N = 3,848). AIREMLF90 and BLUPF90 were used for variance components estimation and EBVs calculation, respectively [24, 25]. In the second step, EBVs as the response variable and SNP genotypes as the

**Table 1. Descriptive statistics of the litter size in Baluchi sheep.**

| Trait | N | Mean | SD | Min | Max | CV |
|---|---|---|---|---|---|---|
| Litter size | 3848 | 1.42 | 0.49 | 1 | 2 | 0.35 |

N, Number of records; SD, Standard deviation; CV, Coefficient of variation.

fixed effects fitted in a GLM model as follows,

$$EBVs = X_{SNP}\beta_{SNP} + PCs_{[1:6]} + e$$

where $X_{SNP}$ is the incidence matrix relating EBVs to SNP genotypes and $\beta_{SNP}$ is the regression coefficient. The analysis was performed using the GenABEL [22] package in the R environment. In this step to account for population stratification, a principal component analysis (PCA) was performed and the first six principal components (based on cumulative variance) were incorporated in the model as covariates. However, after using PCs as covariates, partial inflation was corrected using the genomic control (GC) method, and all p-values were presented without any inflation (λ = 1). For multiple testing corrections, the simpleM [26] approach was used. This method works based on the effective number of independent markers (tests). Based on the average number of independent tests on each chromosome and the 0.05 p-value cutoff, we determined a chromosome-wide threshold (0.05/495 = $1.01^*10^{-4}$). Briefly, The SimpleM first computes the eigenvalues from the pair-wise SNP correlation matrix created with composite LD from the SNP dataset and then infers the effective number of independent tests (Meff_G) through principle component analysis. Once Meff_G is estimated, a standard Bonferroni correction is applied to control for the multiple testing. The Manhattan plot was drawn by CMplot (https://github.com/YinLiLin/R-CMplot) R package.

Two well-known databases including BioMart-Ensembl (www.ensembl.org/biomart) and UCSC Genome Browser (http://genome.ucsc.edu) were used along with the Ovis aries reference genome assembly (Oar_v3.1) to identify genes.

## Genotypes mean comparison

To investigate the genotype effect of significant SNPs on litter size, differences between each genotype means were analyzed by Tukey's multiple pairwise-comparison test using R. The p-values less than 0.05 ($p < 0.05$) were considered statistically significant. The ggplot2 [27] R package was used to visualize the results.

## Gene-set analysis

Gene-set analysis is usually done in 3 steps [17]. In the first step, an arbitrary threshold of p-value $\leq 0.05$ was used to determine significant SNPs [17, 28]. The biomaRt [29, 30] R package and the Oar_v3.1 ovine reference genome assembly were used to assign SNPs to genes if they were located within the genes or 15 kb upstream or downstream of a gene. The size of the flanking region was based on the finding that most SNPs that affect the expression of genes are located within 15 kb of the gene [31]. Genes with at least one significant SNP were considered as significantly associated genes. In the second step, Gene Ontology (GO) and Kyoto Encyclopedia of Genes and Genomes (KEGG) databases were used to define the functional gene-sets. And finally, the significant association of a particular gene-set with the litter size was calculated using Fisher's exact test based on the hypergeometric test [17]. The p-value of g significant genes in the gene-set was computed by,

$$P - value = 1 - \sum_{i=0}^{g-1} \frac{\binom{s}{i}\binom{m-s}{k-i}}{\binom{m}{k}}$$

where $s$ is the total number of significant genes associated with the litter size, $m$ is the total number of analyzed genes, and k is the total number of genes in the gene-set under consideration

[17]. Both GO and KEGG enrichment analyses were performed using the R package clusterPro-filer [32]. Gene-sets with more than 5 and less than 500 genes were tested. Categories with a p-value less than or equal to 0.05 (P ≤ 0.05) were considered significant sets. The ggplot2 [27] R package was used to visualize the GO and KEGG analysis results as bar plots.

## Results

### Estimates of genetic parameters

Estimates of variance components, heritability ($h^2$), repeatability (r), and range of estimated breeding values in genotyped ewes are shown in Table 2. The estimate of heritability of the litter size trait was 0.08 which was in the range reported by previous authors [33–35] and as expected, it was low.

### Genome-wide association analysis

In this study, a two-step approach was applied to perform a GWA analysis to identify gene variants affecting litter size in Baluchi sheep. We used EBVs as the response variable for GWAS which can increase the power to some extent as we have a better estimate of the actual genetic variance. EBVs can largely compensate for the limited number of genotypes to get reasonable estimates. After quality control, 84 ewes and 45342 SNPs were retained for the GWA analysis. We applied the simpleM method for multiple testing correction. This approach can take into account the dependence among the SNPs that are in linkage disequilibrium (LD) and consequently a less stringent threshold can be applied. Although false discovery tests are important for controlling false positives, if there are numerous SNPs in strong LD, they could cause substantial loss of statistical power and increase risks of missing true associations (false negatives) [36]. In addition, we used both the kinship matrix (K) and PC levels (Q) for considering family relatedness and population structure. Previous studies [37–39] have debated that Bonferroni correction for marker effects using both Q and K could result in over-correcting and the need to use a lower significance level of the p-value. The results of GWA analysis are shown in the Manhattan plot in Fig 1.

Three SNPs located on OAR2, OAR10, and OAR25 were identified as chromosome-wide significantly associated with litter size in Baluchi sheep (Table 3).

The Pairwise-comparisons of significant SNP genotypes are shown in Fig 2. The significant SNP, rs406187720, on OAR2, was located within the *NTRK2* gene. For this SNP, EBV for A/A (n = 1) genotype was greater than the mean EBV for G/A (n = 18) and G/G (n = 64) genotypes and the difference between G/A and G/G was statistically significant (p < 0.05). Mean comparisons (Fig 2) and the regression coefficient of A allele and A/A genotype (Table 2) of this SNP showed an additive effect. The SNP on OAR10 was not near any gene in 1Mb span, but genotypes mean comparison of this SNP showed that the addition of an A allele could increase the EBV for ewes carrying that allele (Fig 2). Another significant SNP, rs430079982, was identified on OAR25 within ovine known gene *RAB4A*. For this SNP, the EBV of ewes with the genotype G/G (n = 44) was significantly greater than that of the ewes with the genotype G/A (n = 31)

**Table 2. Estimates of variance components, genetic parameters, and range of estimated breeding values.**

| Trait | $\sigma_p^2$ | $\sigma_a^2$ | $\sigma_{pe}^2$ | $\sigma_e^2$ | $h^2 \pm SE$ | $r \pm SE$ | EBVs |
|---|---|---|---|---|---|---|---|
| Litter size | 0.181 | 0.015 | 0.036 | 0.130 | 0.083±0.027 | 0.282±0.015 | -0.122 to +0.161 |

$\sigma_p^2$, Phenotypic variance; $\sigma_a^2$, Additive genetic variance; $\sigma_{pe}^2$, Permanent environmental variance; $\sigma_e^2$, Residual variance; $h^2$, Heritability; r, Repeatability; SE, Standard error; EBVs, Estimated breeding values range for genotyped animals.

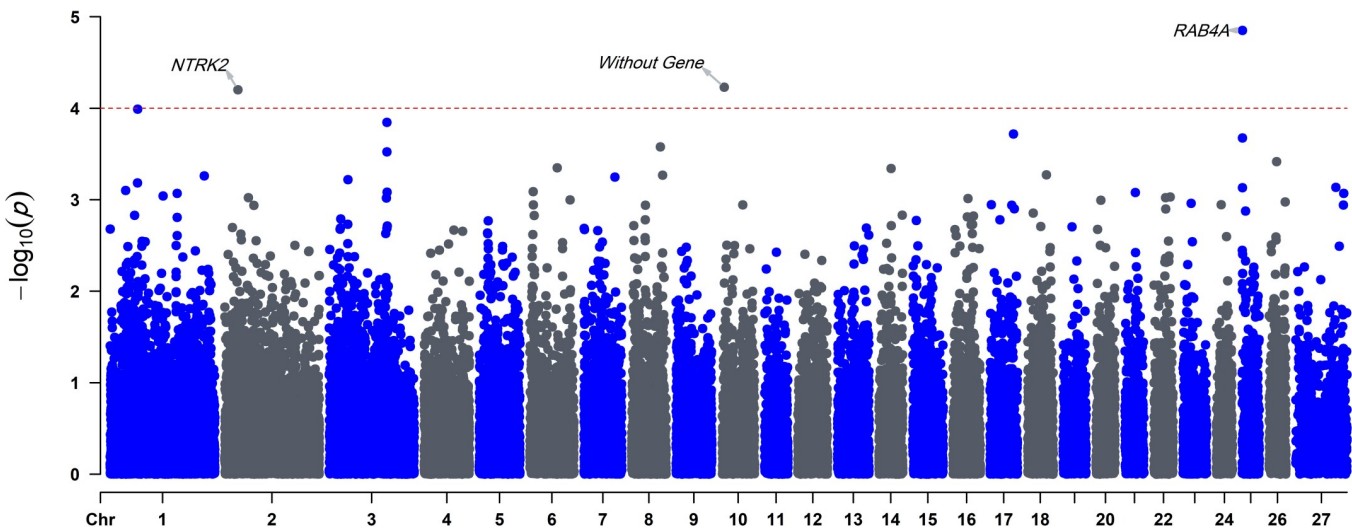

**Fig 1. Manhattan plot for associations of SNPs with the litter size in Baluchi sheep.** X-axis: SNPs positions on chromosomes, Y-axis: -Log₁₀ p-value. The red dashed line indicates the threshold for chromosome-wide ($P<1.01*10^{-4}$) statistical significance based on the simpleM approach. All p-values were presented without any inflation ( = 1).

and A/A (n = 9) (p < 0.01 and p < 0.001, respectively) (Fig 2). As it is clear from regression coefficients in Table 2, this SNP has a negative effect on litter size.

## Gene set analysis

The genes or genomic regions that were identified in GWAS explain only part of the genetic variation. To overcome this limitation, the GWAS was complemented with a gene set analysis using the GO and KEGG database to detect potential functional categories underlying the litter size. In total, from 45,342 SNPs tested in the GWAS, 23,462 SNPs were located within or 15 kb upstream or downstream of 15,815 genes in the Oar.v3.1 ovine genome assembly. About 1,235 genes, out of 15,815 genes contained at least one significant SNP (p-value ≤ 0.05) and defined as significantly associated genes with the litter size. GO terms and KEGG pathways with a nominal p-value ≤ 0.05 were reported as significant (Fig 3).

Two KEGG terms related to steroid hormones, including *Ovarian steroidogenesis* (oas04913) and *Steroid hormone biosynthesis* (oas00140) showed an overrepresentation of significant genes associated with litter size. Many KEGG terms related to signaling also were identified as significant terms including *Calcium signaling pathway* (oas04020), *Sphingolipid signaling pathway* (oas04071), *Chemokine signaling pathway* (oas04062), and *Phospholipase D*

**Table 3. Significantly (chromosome-wide) associated SNPs with the litter size in Baluchi sheep.**

| Trait | Chr | SNP | Location | $\beta_{allele}$[a] | $\beta_{Genotype}$[b] | Closest gene | Distance (bp) | Adjusted p-value (GC[c]) |
|-------|-----|-----|----------|---------|-----------|--------------|---------------|--------------------------|
| Litter size | 25 | rs430079982 | 620434 | -0.05 | -0.09 | RAB4A | Within | $1.40×10^{-5}$ |
| | 10 | rs407696726 | 6730970 | 0.05 | 0.06 | Without gene | 1Mb | $5.88×10^{-5}$ |
| | 2 | rs406187720 | 34802714 | 0.07 | 0.14 | NTRK2 | Within | $6.28×10^{-5}$ |

Chr, Chromosome;

[a] Regression coefficient for the effective (minor) allele (A);

[b] Regression coefficient for effective allele homozygote genotype (A/A);

[c] Genomic control.

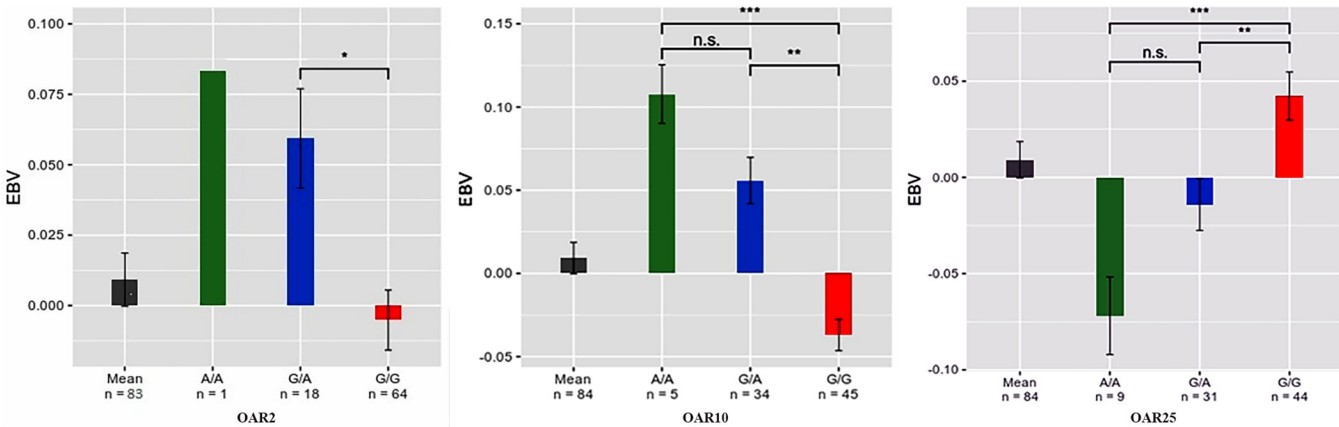

**Fig 2. Pairwise-comparisons of the significant SNP genotypes for the litter size in Baluchi sheep.** Differences between EBV means of each genotype in each significant SNP were calculated. Tukey's multiple pairwise-comparison test was used for mean comparisons between genotypes. For significant SNP on OAR2, mean comparison only performed between G/A and G/G genotypes. *: $p < 0.05$, **: $p < 0.01$, ***: $p < 0.001$, n.s: not significant.

*signaling pathway* (oas04072). Also, we identified several GO terms associated with litter size. Pathways related to the adhesion process e.g. *biological adhesion* (GO:0022610) and *cell adhesion* (GO:0007155) were among the significant terms. The *neuromuscular junction* (GO:0031594), *cell junction* (GO:0030054), and *cell-cell adherens junction* (GO:0005913) were other GO terms that enriched with significant genes. Two significant GO terms including *ion channel regulator activity* (GO:0099106) and *calcium channel regulator activity* (GO:0005246) were related to ion transport and channel activity. The *cell cycle G2/M phase transition* (GO:0044839) was another GO term which overrepresented with significant genes.

## Discussion

### Genome-wide association analysis

In the present study, to identify genomic regions and candidate genes associated with the litter size in sheep, a whole-genome scan was performed. Also, to cover some limitations of GWAS,

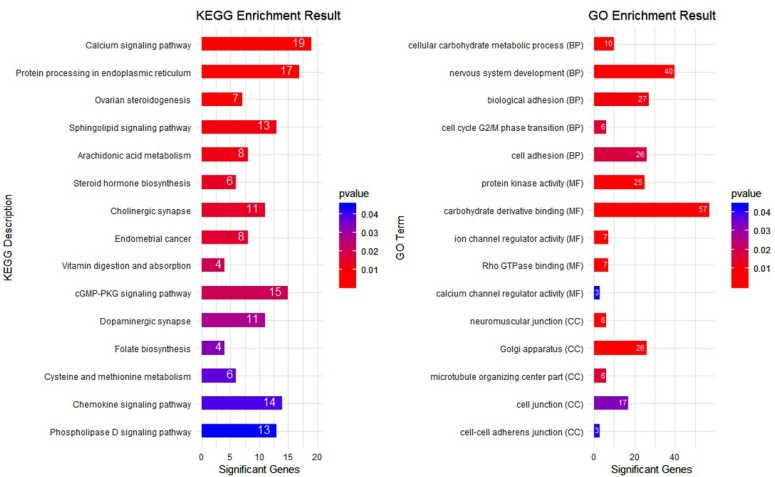

**Fig 3. Top Gene Ontology (GO) terms and KEGG descriptions significantly (p ≤ 0.05) enriched using genes associated with litter size in Baluchi sheep.** The numbers within the bars denote the number of significant genes inside GO/KEGG terms.

a gene set analysis was used as a complementary approach. We identified the Neurotrophic receptor tyrosine kinase 2 (*NTRK2*) gene on OAR2 as a candidate gene affecting litter size in sheep. As shown in Fig 2, in the *NTRK2* gene, the addition of an A allele increases the average EBV in the A/G genotype, and the difference between the A/G and G/G genotypes was significant. Although the individual with A/A genotype has the greatest EBV than the other two genotypes, due to a single observation, the result needs to be confirmed by larger sample size. Of course, it should be noted that the EBVs are calculated not only by the individual records but also by using their relatives' records and means that contain information from both the individuals and relatives of the individuals. However, the *NTRK2* gene functions in oocyte development and survival, make the result sensible.

Neurotrophic receptor tyrosine kinase 2 (*NTRK2*) is a transmembrane receptor that binds to brain-derived neurotrophic factor (*BDNF*) and neurotropin-4/5 (*NTF-4/5*) with high affinity. The *BDNF* and *NTF-4/5* are members of neurotrophins (NTs). It has long been thought that NTs and their receptors to be exclusively required for the development of the nervous system. But, findings suggest that they are vital for the control of ovarian functions and contribute to both the formation and development of follicles [40]. Expression of *BDNF* and *NTRK2* during the preimplantation have been observed in the oviduct, trophectoderm cells, and uterus in mice [41]. Also, their expression in oviducts and uteri of nonpregnant sheep and high expression in uteri of pregnant sheep has been reported [42]. Studies using *Ntrk2*-null mice showed that this molecule work as a receptor which is essential for follicular assembly, structural integrity and, early follicular growth [43, 44]. It has been shown that primordial follicle formation is decreased in the absence of *NTRK2* or its ligands, *NTF4/5*, and *BDNF*. This deficiency just takes place after the initiation of folliculogenesis in the absence of the *NTRK2* receptor. The ligand-mediated activation of the *NTRK2* receptor enhances the capacity of the infantile ovary to respond to FSH with the synthesis of cyclin D2 which is a cell cycle protein mediating the proliferative actions of FSH in the ovary [45]. Both the number of secondary follicles and FSH receptor (*FSHR*) expression are reduced in *Ntrk2*-null ovaries and suggested that *NTRK2* facilitate subsequent follicle development by inducing the formation of functional FSH receptors [46].

It has been shown that the preovulatory surge of gonadotropins increases *BDNF* synthesis in granulosa cells which facilitate oocyte development by mediating *NTRK2* receptors expressed in the oocyte. Consequently, this process supports the development of zygotes into preimplantation embryos by PI3K-AKT signaling pathway activation [41, 47]. Oocytes without the *NTRK2* gene can't respond to gonadotropins due to a lack of PI3K-AKT mediated signaling activation [48]. Notably, the *PI3K-Akt signaling* pathway (oas04151) was one of our identified KEGG pathways with 21 significant genes, but it was not significant overall. Recently, using RNA-seq analysis, the PI3K-Akt signaling pathway was reported as a significant KEGG term affecting prolificacy [49].

It has been shown that kisspeptin and its receptor, *KISS1R* are vital for *NTRK2* function, and oocytes lacking *KISS1R* do not respond to gonadotropins while NTRK2 expression is normal [50]. According to the model described by Dorfman et al. [50], before puberty, oocytes only express a truncated *NTRK2* form (*NTRK2*.T1) receptors and *KISS1R*. The preovulatory surge of gonadotropins at puberty increases the production of BDNF and kisspeptin in granulosa cells which binds to their receptors (*NTKR2*.T1 and *KISS1R*) in oocytes and promote the development of full-length and active *NTRK2* receptors (*NTKR2*.FL) that activate a PI3K-AKT-mediated pathway which ensures oocyte survival. In lack of full-length *NTRK2* (*NTRK2*.FL) the follicles lose their integrity and the oocytes can't survive [50]. This suggested model by Dorfman et al. [50] to explain the *NTRK2* mechanism of action can be seen here (https://academic.oup.com/view-large/figure/62307720/zee9991476020008.tif). NTRK2.FL: Full-length NTRK2; NTRK2.T1: Truncated NTRK2.

To date, most reports of the *NTRK2* gene have been limited to molecular studies, but in a recent study, using RNA-seq analysis, Xia et al. [51], tried to identify fecundity-related lncRNAs and mRNAs in Han sheep. Notably, they identified both 105,603,287 lncRNA and its target gene, *NTRK2*, as significantly upregulated genes associated with sheep prolificacy. In addition, they reported *Ovarian steroidogenesis* and *Steroid hormone biosynthesis* KEGG pathways as significant, which both of them were identified in this study. In another study, the *NTRK2* was found to be differentially upregulated in the ovaries of Hu sheep in association with off-season reproduction [52]. Besides, it's reported that a variant close to the *NTRK2* gene are associated with birth weight in female twins in human [53]. Based on the results of this study and other studies mentioned above, the *NTRK2* gene can be introduced as an important candidate for further studies related to sheep prolificacy.

The SNP on OAR10 was identified as significant but was not near any gene in the 1Mb span. Genotypes mean comparison showed that the addition of A allele could significantly increase the average of EBV in A/G and A/A genotypes in comparison with the G/G genotype. However, the difference between G/A and A/A genotype was not significant (Fig 2).

Another significant SNP was rs430079982 located within the *RAB4A* gene on OAR25. This gene encodes Ras-related protein Rab-4A in humans. *RAB4A* is a member of the Rab GTPase superfamily which has many roles in membrane trafficking. Also, this gene regulates receptor recycling from endocytic compartments to the plasma membrane [54, 55]. *RAB4A* was firstly identified in early endosomes in Chinese hamster ovary [56]. It has been reported that *RAB4A* has a role in endosome-to-plasma membrane recycling of vascular endothelial growth factor receptor 2 (*VEGFR-2*) which is vital for blood vessel formation, cell migration, intracellular signaling, and proliferation through angiogenesis [57]. Angiogenesis is linked with follicular development and is controlled independently within each follicle which potentially making the functioning of its vasculature critically important in determining its fate [58]. Establishment and remodeling of a complex vascular system enable the follicle and corpus luteum to receive the required supply of nutrients, oxygen, and hormonal support as well as facilitating the release of steroids [59]. It has been reported that the production and action of vascular endothelial growth factor A (*VEGFA*) are necessary for follicular growth, ovulation, and the development and function of the corpus luteum [60]. *VEGF* is a potent and specific stimulator of vascular endothelial cell proliferation acting through two tyrosine kinase receptors, *VEGFR-1* and *VEGFR-2* [58]. Administration of *VEGFA*, stimulate the development of secondary follicles in cows [60]. Inhibition of *VEGF* and its receptor *VEGFR-2* can inhibit follicular development or prevent ovulation [58]. Genotypes mean comparison showed that the average EBV for G/G genotype was significantly different from G/A and A/A genotypes.

## Gene set analysis

In this study, the GWAS was complemented with a gene-set analysis using KEGG and GO databases to detect potential functional categories underlying the litter size. A full list of significantly enriched pathways using GO and KEGG databases are presented in S1–S3 Tables. Notably, two KEGG pathways related to Steroid hormone including *Ovarian steroidogenesis* (oas04913) and *Steroid hormone biosynthesis* (oas00140) were significant. Recently, both pathways were identified in a transcriptome analysis in association with prolificacy in Han sheep [51]. Also, using protein profile analysis, both *Ovarian steroidogenesis* and *Steroid hormone biosynthesis* were reported as significant KEGG pathways affecting prolificacy [61]. In addition, they identified the *protein digestion and absorption* KEGG pathways as significant. Similarly, in this study, the *Protein processing in endoplasmic reticulum* (oas04141) KEGG term was identified as a pathway associated with protein metabolism. Both *Ovarian steroidogenesis* and

*protein digestion and absorption* KEGG pathways have recently been reported in a GWAS study as pathways affecting prolificacy in sheep [62]. Many signaling KEGG pathways were identified and some of them were related to reproductive processes. The *calcium signaling pathway* as the most significant term was one of them. Calcium has an important function in the oocyte's meiotic maturation. Recently, Hernández-Montiel et al. [63] using RNA-seq transcriptome analysis in prolific and non-prolific Mexican Pelibuey sheep breed, reported the *Calcium signaling pathway* as a significant KEGG term. In addition, Xu et al. [64] in a GWAS, identified *calcium ion binding* GO term as pathway affecting prolificacy in Chinese sheep. Notably, we identified *calcium channel regulator activity* (GO:0005246) as a significant GO term in our gene set analysis using the GO database.

The *Chemokine signaling* (oas04062) was another significant KEGG pathway. Different aspects of ovulation are very similar to the inflammatory process and a diverse set of leukocytes have been observed in different stages of follicle development and ovulation [65]. *Phospholipase D signalling* (oas04072) was another significant KEGG pathway in the present study. Banno et al. [66] suggested that Phospholipase D (PLD) has a role in growth factor regulation in at least two signaling pathways, including extracellular signal-regulated kinases (ERKs) and the PI3k-Akt pathway. In addition, PLD is involved in the inflammatory process. IL-8 as an inflammatory cytokine up-regulates the expression of PLD and enables PLD to facilitate the migration of immune cells [67]. Notably, we identified two functionally related KEGG pathways including *Folate biosynthesis* (oas00790) and *Cysteine and methionine metabolism* (oas00270). Folate (B9 vitamin) is essential for rapid cell growth and division, which usually occurs during follicle development and fetal growth. Folate deficiency can lead to homocysteine accumulation. Homocysteine originates from the methionine and is remethylated to methionine with folates acting as methyl donors [68]. Adverse impacts of homocysteine accumulation due to folate deficiency on female reproductive performance can include inflammatory cytokine production, low cell division [69], increase in apoptosis [70], and high oxidative stress [71] which affects many aspects of oocyte development. Women who take folic acid supplements have lower homocysteine concentrations in their follicular fluid, oocytes with better quality, and a high degree of mature oocytes [72]. It has been reported that the folate concentration in prolific breeds ewes is higher than from non-prolific breeds [73], but a more recent study reported that folic acid supplementation to prolific and nonprolific ewes increase folate concentration in red cells and plasma but hasn't any effect on fertility or litter size [74].

Our gene set analysis using the GO database identified several terms associated with litter size. Two pathways related to carbohydrate metabolism including *cellular carbohydrate metabolic process* (GO:0044262) and *carbohydrate derivative binding* (GO:0097367) showed an overrepresentation of significant genes. Carbohydrate and lipid oxidation have a major impact on fecundity. Glucose and pyruvate are the main energy source for mouse ovarian follicles and oocyte, respectively. Using in vitro grown follicles, it has been reported that glucose uptake and lactate production by follicles are increased parallelly with development and are stimulated just before ovulation [75]. In another study ovarian proteome of hyper-prolificacy sheep was analyzed and more than 100 identified proteins were classified in carbohydrate transport and metabolism [76]. Also, Yang et al. [77] using transcriptome analysis, reported *galactose metabolism* and *fructose and mannose metabolism* as biological pathways affecting ovulation. Many GO terms related to cell communication such as *biological adhesion* (GO:0022610), *cell adhesion* (GO:0007155), *cell junction* (GO:0030054), and *cell-cell adherens junction* (GO:0005913) were identified. Given that the oocytes are surrounded by many cell layers, it makes sense that signaling and communication between the oocytes and the environment have a great impact on follicle growth and the ovulation process. Besides the other communication ways between the oocyte and surrounding cells, physical connections are essential for follicular growth and oocyte maturation. Cadherin family genes

as part of the *cell adhesion* process, encode glycoproteins that are responsible for cell-cell communication through calcium ions [78]. The *gap junction* is one of the direct child terms of *cell junction* pathway which contains several proteins called connexin. These proteins form intercellular membrane channels that connect adjacent cells and allow for direct sharing of several metabolites such as glucose, amino acids, nucleotides, ions, and second messengers (e.g. cGMP) to support the oocyte development, metabolism, and hemostasis [79]. Many genomics, transcriptomics, and proteomics studies reported many pathways related to both cell adhesion and gap junction process as pathways which has vital roles in prolificacy [49, 62, 77, 80, 81].

It should be noted that, at this point, reported genes are just candidates. The identification of validated causal genetic variants that underlie reproduction traits is big challenges in livestock genetic research and methods such as whole-genome resequencing, high-throughput transcriptome sequencing, and functional studies are required to fully address the causal relationship between genetic variants and phenotypes [82, 83]. Therefore, although there are several reports about the *NTRK2* gene functions in oocyte growth, development, and survival, its variant's effect on the litter size needs to be confirmed with a larger sample size. Recently, however, its effect on sheep prolificacy has been reported using transcriptome analysis [51]. Finally, while this study does improve our understanding of an interesting but less characterized breed, it will still be useful to see if these results can be broadly applicable to other breeds as well.

## Conclusions

In the present study, a two-step GWAS accompanied by a gene set analysis (GSA) as a complementary approach was utilized to identify genes and pathways affecting the litter size in Iranian Baluchi sheep. Two new genes including *NTRK2* and *RAB4A* and one genomic region on OAR10 were identified as regions associated with the litter size in sheep. The *NTRK2* gene product works as a receptor which is essential for follicular assembly, early follicular growth, and oocyte survival by activation of the PI3K-AKT signaling pathway. This is the first study that reports the *NTRK2* gene affecting litter size in sheep. Another identified gene, *RAB4A*, is involved in intracellular signaling and proliferation through angiogenesis. Several KEGG pathways related to steroid hormone biosynthesis and signaling were identified. In addition, we identified many GO terms related to carbohydrate metabolism and cell communication via adhesion and junctions as significant terms. Finally, reported genes and pathways provide valuable information that can be helpful to a better understanding of the mechanisms underlying sheep litter size variation.

## Supporting information

**S1 Table. GO terms.** GO terms significantly (P $\leq$ 0.05) enriched using genes associated with litter size in Baluchi sheep.
(XLSX)

**S2 Table. KEGG pathways.** KEGG pathways significantly (P $\leq$ 0.05) enriched using genes associated with litter size in Baluchi sheep.
(XLSX)

**S3 Table. GWAS results.** GWAS results for litter size in Baluchi sheep.
(XLSX)

## Acknowledgments

SMEF thanks the Abureyhan Campus of the University of Tehran for hosting during the research.

## Author Contributions

**Conceptualization:** Seyed Mehdi Esmaeili-Fard.

**Data curation:** Seyed Mehdi Esmaeili-Fard.

**Formal analysis:** Seyed Mehdi Esmaeili-Fard.

**Methodology:** Seyed Mehdi Esmaeili-Fard, Mohsen Gholizadeh, Rostam Abdollahi-Arpanahi.

**Project administration:** Seyed Hasan Hafezian.

**Resources:** Mohsen Gholizadeh.

**Software:** Seyed Mehdi Esmaeili-Fard.

**Supervision:** Mohsen Gholizadeh, Seyed Hasan Hafezian, Rostam Abdollahi-Arpanahi.

**Visualization:** Seyed Mehdi Esmaeili-Fard.

**Writing – original draft:** Seyed Mehdi Esmaeili-Fard.

**Writing – review & editing:** Seyed Mehdi Esmaeili-Fard, Mohsen Gholizadeh, Seyed Hasan Hafezian, Rostam Abdollahi-Arpanahi.

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
