## [Decision Letter · Decision Letter 0]

5 Oct 2020

PONE-D-20-25223

NTRK2 and litter size in sheep: an insight from GWAS and pathway enrichment analysis

PLOS ONE

Dear Dr. Esmaeilifard,

Thank you for submitting your manuscript to PLOS ONE. After careful consideration, we feel that it has merit but does not fully meet PLOS ONE’s publication criteria as it currently stands. Therefore, we invite you to submit a revised version of the manuscript that addresses the points raised during the review process.

The idea of the article is good. But the study needs to provide detailed supportive data to validate the conclusion drawn. The Introduction section needs to be edited carefully for cohesiveness and flow. Further the title should be edited and it should reflect the actual results obtained.

We look forward to receiving your revised manuscript.

Kind regards,

Academic Editor

PLOS ONE

Journal Requirements:

2.We note that Figure(s) [4] in your submission contain copyrighted images. All PLOS content is published under the Creative Commons Attribution License (CC BY 4.0), which means that the manuscript, images, and Supporting Information files will be freely available online, and any third party is permitted to access, download, copy, distribute, and use these materials in any way, even commercially, with proper attribution. For more information, see our copyright guidelines: http://journals.plos.org/plosone/s/licenses-and-copyright.

1.    You may seek permission from the original copyright holder of Figure(s) [4] to publish the content specifically under the CC BY 4.0 license.

3.We note that you have stated that you will provide repository information for your data at acceptance. Should your manuscript be accepted for publication, we will hold it until you provide the relevant accession numbers or DOIs necessary to access your data. If you wish to make changes to your Data Availability statement, please describe these changes in your cover letter and we will update your Data Availability statement to reflect the information you provide.

4. Please include a copy of Table 3 which you refer to in your text on page 7.

Additional Editor Comments (if provided):

The idea of the article is good. But the study needs to provide detailed supportive data to validate the conclusion drawn. The Introduction section needs to be edited carefully for cohesiveness and flow. Further the title should be edited and it should reflect the actual results obtained.

Reviewers' comments:

Reviewer's Responses to Questions

**Comments to the Author**

1. Is the manuscript technically sound, and do the data support the conclusions?

Reviewer #1: Partly

Reviewer #2: No

Reviewer #3: Partly

2. Has the statistical analysis been performed appropriately and rigorously? 

Reviewer #1: No

Reviewer #2: No

Reviewer #3: Yes

3. Have the authors made all data underlying the findings in their manuscript fully available?

Reviewer #1: No

Reviewer #2: No

Reviewer #3: Yes

4. Is the manuscript presented in an intelligible fashion and written in standard English?

Reviewer #1: Yes

Reviewer #2: No

Reviewer #3: Yes

5. Review Comments to the Author

Reviewer #1: Reviewer Recommendation and Comments for Manuscript Number PONE-D-20-25223

• The manuscript is acceptable to be published in PLoS One with the below given modifications and additions.

Major comments:

1) The manuscript is well written and researched. Specifically, I would appreciate the authors for trying find out the genes responsible for litter size in sheep but the information provided in the manuscript and the conclusions drawn are too ambitious. There is a need for supporting tests and data, for the drawn conclusions. The conclusions need evidences through minimal wet lab experiments. The supportive data is missing.

2) The author should cite and include more data regarding NTRK2 to support their study.

Minor comments:

1. Please provide justification for the tests used in materials and methods section. Proper justification through citation is missing.

2. As it is mentioned (page 3, lines-54-55) that a previous study by Gholizadeh et al. (2014) identified two significant SNP (rs407696726 and rs412433416) on OAR10 and OAR15 in association with litter size, and in conclusion (page 15, lines-311-312) authors also claimed that they also found a one genomic region on OAR10 associated with the litter size in sheep, how this study is different or novel from the Gholizadeh et al. study?

3. Page 11, lines- 225-226, the citation number given in fig 4 doesn’t match with the reference list. Correction required.

4. Manuscript can be improved a lot in terms of content to be acceptable in Plos one.

Reviewer #2: In the manuscript, the authors described the identification of the gene variants and pathways and provides important findings which can be useful for better understanding of the mechanisms underlying in iranian baluchi sheep litter size variation.

The major concern with the manuscript is the lack of any experimental data that shows NTRK2 gene involvement in affecting the litter size in sheep. I think the authors should consider including some experimental results in the revised draft.

The other major issue with the manuscript that the title did not correlate with the purposed findings. The title must can be changed to the appropriate one.

The introduction section seems incomplete, should be re-written to provide more clear background of the study.

The methodology used for the analysis seems appropriate however, there is no detailed description of the methods for example GO and KEGG methodology should be mentioned as a different section in the Materials and Methods.

Typo-Error:

1. Page 7 line 139 "Table 3" changed it into "Table 2"

2. Page 14 line 301-302 Re-write the line " The cell junction.........connexins"

Reviewer #3: Mehdi Esmaeilifard etal, in the entitled “ NTRK2 and litter size in sheep: an insight from GWAS and pathway enrichment” manuscript, investigated the genetic mechanism that affects litter size in Iranian Baluchi Sheep. It’s indeed an interesting study to discover what genetic factor and its mechanism that contributes to litter size. By employing two-step genome wide association study and gene set enrichment analysis (with GO and KEGG databases), authors intriguingly identified two genes; namely NTRK2 and RAB4A that are associated with litter size in sheep. The function of NTRK2 in litter size homeostasis is significantly supported by the given data. Additionally, the known function of NTRK2 in oocyte development and survival process provides a direct link for NTRK2 in affecting litter size. Overall the here identified NTRK2 role in litter size is clear.

Major point

Authors showed that the second gene RAB4A significantly affects litter size, however, its known role in membrane trafficking, blood vessel formation and proliferation makes it hard to corelate RAB4A function in affecting litter size. It could be that the observed phenotype of RAB4A on litter size is an indirect effect. It is essential to explain how RAB4A is influencing litter size?

Specific point

Authors should carefully check the cited references throughout the manuscript. For example, in figure 4 the author cited Dorfman etal (28), but actually in the reference list the correct number is 29. Additionally, it's nice if author follows a standard reference style, either number or name. For example Gholizadeh et al. (2014) is cited in the text.

Figure 2: A/A has greater EBV for litter size ( n=1) than the other two A/G and G/G genotypes, it's good to show larger sample size for A/A group for statistical purpose.

Minor points

It would be good if author gives explanation of their abbreviations in the manuscript for example; EBV, SNP etc..

6. PLOS authors have the option to publish the peer review history of their article (what does this mean?). If published, this will include your full peer review and any attached files.

Reviewer #1: No

Reviewer #2: No

Reviewer #3: No

---

## [Author Response · Author response to Decision Letter 0]

19 Nov 2020

Dear Editor,

I am pleased to resubmit the revised version of our Manuscript: previous title: “NTRK2 and litter size in sheep: an insight from GWAS and pathway enrichment analysis"; new title: “Genome-wide association study and pathway analysis identify NTRK2 as a novel candidate gene for litter size in sheep” for publication in PLOS ONE journal. We would like to thank the reviewers for the important criticism that they raised here. The manuscript revised point by point and changes are highlighted with track changes. The line numbers of the added lines to the manuscript are mentioned below the comments whenever was needed.

Also, the datasets supporting the conclusions of this article are publicly available in the Figshare repository under the following DOI number: https://doi.org/10.6084/m9.figshare.12515189.v1

Sincerely,

Mehdi Esmaeilifard

###

Response for Reviewers' comments:

The authors would like to thank the reviewers for helpful comments and technical corrections. 

Reviewer 1

The manuscript is acceptable to be published in PLoS One with the below given modifications and additions.

Major comments:

1- The manuscript is well written and researched. Specifically, I would appreciate the authors for trying find out the genes responsible for litter size in sheep but the information provided in the manuscript and the conclusions drawn are too ambitious. There is a need for supporting tests and data, for the draw conclusions. The conclusions need evidences through minimal wet lab experiments. The supportive data is missing.

AU: We totally agree with the referee that the lack of a validation study is the limitation of this study. But as you know, most GWAS studies which performed up to now, just reported the results and did not consider any validation study. Validation of identified causal genetic variants that underlie the traits is the main challenge of current research. Costly and time-consuming methods such as whole-genome resequencing, high-throughput transcriptome sequencing, and functional studies are required to fully unravel the causal relationship between genetic variants and phenotypes. Since we do not have wet lab now, it is not possible for us to perform such a validation study. However, we complemented our study using Gene Set Analysis (GSA) to see can these identified genes, enrich pathways related to ovarian functions or not? We could identify many closely related pathways that were reported many times in other studies as significant pathways associated with prolificacy. 

As an alternative, we could pretty rely on the same reports in other studies, especially those that were reported in transcriptome analysis. During the revise, we found a recently published study, in which using RNA-seq analysis, Xia et al. (2020), tried to identify fecundity-related lncRNAs and mRNAs in Han sheep. Notably, they identified both 105,603,287 lncRNA and its target gene, NTRK2, as significantly upregulated genes associated with Han sheep prolificacy. In addition, they reported Ovarian steroidogenesis (oas04913) and Steroid hormone biosynthesis (oas00140) KEGG pathways as significant which both of them were identified in this study. We added Xia et al. (2020) study results to the revised version of the manuscript.

We also added a paragraph to the end of dissection section to provide a more in-depth discussion of the caution associated with making inferences. 

Lines 306-316 and 417-426 were added to the manuscript to clarify this part.

2- The author should cite and include more data regarding NTRK2 to support their study.

AU: To date, most reports of the NTRK2 gene have been limited to molecular studies, but fortunately during the revise, we found a recently published study, in which using RNA-seq analysis, Xia et al. (2020), tried to identify fecundity-related lncRNAs and mRNAs in Han sheep. Notably, they identified both 105,603,287 lncRNA and its target gene, NTRK2, as significantly upregulated genes associated with sheep prolificacy. We added Xia et al. (2020) study results to the revised version of the manuscript (Lines 306-311). Also, two more reports were added (Lines 311-216). In addition, two paragraphs about the NTRK2 role and mechanism were added (Lines 269-273 and 280-287). 

Minor comments:

1- Please provide justification for the tests used in materials and methods section. Proper justification through citation is missing.

AU: Corrected as suggested. Many details and citations were added.

2- As it is mentioned (page 3, lines-54-55) that a previous study by Gholizadeh et al. (2014) identified two significant SNP (rs407696726 and rs412433416) on OAR10 and OAR15 in association with litter size, and in conclusion (page 15, lines-311-312) authors also claimed that they also found a one genomic region on OAR10 associated with the litter size in sheep, how this study is different or novel from the Gholizadeh et al. study?

AU: Our study has many main differences with Gholizadeh et al. study. We followed a two-step approach: first, we combined all parities and performed the analysis in one run. So, we could use the repeatability model (a kind of mixed model which usually are used for repeated measurements) and could consider parity effect, dam age, and birth year effect, we also could use the relationship matrix (K) and do control for the family relatedness. We used about 430 lambing records in one run and calculate the EBVs for all dams. Using EBVs as a response variable for GWAS can increase the power to some extent as we have a better estimate of the actual genetic variance. In the second step, we used EBVs as the response variable along with the SNPs and fit a model. We also incorporated the first six PC levels (Q) to considering population structure at this step.

Gholizadeh et al. performed the analysis for all parities separately (one analysis for each parity), so they couldn’t consider parity effect, dam age, or birth year. They used a GLM model in plink, so couldn’t consider a K matrix and correct for family relatedness and also didn’t consider PC levels (Q). They didn’t use EBVs as the response variable. 

3- Page 11, lines- 225-226, the citation number given in fig 4 doesn’t match with the reference list. Correction required. [Lines in revised manuscript with track changes: 320-322]

AU: Thanks for the comment. Corrected.

4- Manuscript can be improved a lot in terms of content to be acceptable in Plos one.

AU: Several details were added to all parts of the manuscript based on the reviewers’ comments and the review of the authors which has greatly improved the manuscript.

##############################################################################

Reviewer 2:

In the manuscript, the authors described the identification of the gene variants and pathways and provides important findings which can be useful for better understanding of the mechanisms underlying in Iranian Baluchi sheep litter size variation.

1- The major concern with the manuscript is the lack of any experimental data that shows NTRK2 gene involvement in affecting the litter size in sheep. I think the authors should consider including some experimental results in the revised draft.

AU: We totally agree with the referee that the lack of a validation study is the limitation of this study. But as you know, most GWAS studies which performed up to now, just reported the results and did not consider any validation study. Validation of identified causal genetic variants that underlie the traits is the main challenge of current research. Costly and time-consuming methods such as whole-genome resequencing, high-throughput transcriptome sequencing, and functional studies are required to fully unravel the causal relationship between genetic variants and phenotypes. Since we do not have wet lab now, it is not possible for us to perform such a validation study. However, we complemented our study using Gene Set Analysis (GSA) to see can these identified genes, enrich pathways related to ovarian functions or not? We could identify many closely related pathways that were reported many times in other studies as significant pathways associated with prolificacy. As an alternative, we could pretty rely on the same reports in other studies, especially those that were reported in transcriptome analysis. To date, most reports of the NTRK2 gene have been limited to molecular studies, but fortunately, during the revise, we found a recently published study, in which using RNA-seq analysis, Xia et al. (2020), tried to identify fecundity-related lncRNAs and mRNAs in Han sheep. Notably, they identified both 105,603,287 lncRNA and its target gene, NTRK2, as significantly upregulated genes associated with Han sheep prolificacy. In addition, they reported Ovarian steroidogenesis (oas04913) and Steroid hormone biosynthesis (oas00140) KEGG pathways as significant which both of them were identified in this study. We added Xia et al. (2020) study results to the revised version of the manuscript.

However, we added a paragraph to the end of dissection section to provide a more in-depth discussion of the caution associated with making inferences.

Lines 306-316 and 417-426 were added to the manuscript to clarify this part.

2- The other major issue with the manuscript that the title did not correlate with the purposed findings. The title must can be changed to the appropriate one.

AU: We changed the title to “Genome-wide association study and pathway analysis identify NTRK2 as a novel candidate gene for litter size in sheep”.

3- The introduction section seems incomplete, should be re-written to provide more clear background of the study.

AU: We did an extensive review of the introduction section. We almost rewrote it. 

4- The methodology used for the analysis seems appropriate, however, there is no detailed description of the methods for example GO and KEGG methodology should be mentioned as a different section in the Materials and Methods.

AU: We added more details to the Materials and Methods section to make it clearer. You can see them throughout this section. About the enrichment using GO and KEGG databases: the methodology is the same for both of them and the only difference is in the use of different databases. First, we defined significant and background genes then used this data for enrichment analysis using the hypergeometric test, once using the GO database and once using the KEGG database. We followed Han and Peñagaricano (2016) approach.

5- Typo-Error:

5-1- Page 7 line 139 "Table 3" changed it into "Table 2". [Line in revised manuscript with track changes: 209]

AU: Thanks for the comment. We added a new table (Table 2) in the revised version, so “Table 3” is correct now.

5-2- Page 14 line 301-302 Re-write the line " The cell junction.........connexins". [Lines in revised manuscript with track changes: 407-408]

AU: Done. Lines 407-408.

##############################################################################

Reviewer 3:

Mehdi Esmaeilifard et al, in the entitled “NTRK2 and litter size in sheep: an insight from GWAS and pathway enrichment” manuscript, investigated the genetic mechanism that affects litter size in Iranian Baluchi Sheep. It’s indeed an interesting study to discover what genetic factor and its mechanism that contributes to litter size. By employing a two-step genome-wide association study and gene set enrichment analysis (with GO and KEGG databases), authors intriguingly identified two genes; namely NTRK2 and RAB4A that are associated with litter size in sheep. The function of NTRK2 in litter size homeostasis is significantly supported by the given data. Additionally, the known function of NTRK2 in oocyte development and survival process provides a direct link for NTRK2 in affecting litter size. Overall, the here identified NTRK2 role in litter size is clear.

Major point:

1- Authors showed that the second gene RAB4A significantly affects litter size, however, its known role in membrane trafficking, blood vessel formation and proliferation makes it hard to corelate RAB4A function in affecting litter size. It could be that the observed phenotype of RAB4A on litter size is an indirect effect. It is essential to explain how RAB4A is influencing litter size?

AU: Lines 334-344 were added to the manuscript to clarify this part.

Specific point:

1- Authors should carefully check the cited references throughout the manuscript. For example, in figure 4 the author cited Dorfman et al, (28), but actually in the reference list the correct number is 29. Additionally, it's nice if author follows a standard reference style, either number or name. For example, Gholizadeh et al. (2014) is cited in the text.

AU: Thanks for the comment. All citations were checked again and corrected whenever was needed. In some few lines in which the author name was mentioned, we followed the journal style (Gholizadeh et al. (2014) >>> Gholizadeh et al. [1]).

2- Figure 2: A/A has greater EBV for litter size (n=1) than the other two A/G and G/G genotypes, it's good to show larger sample size for A/A group for statistical purpose.

AU: We appreciate for your comment. The low number of Animals with AA genotype is a limitation of this study, but the individual with A/A genotype had the greatest EBV than the other two genotypes. The EBVs are calculated not only by the individual records but also by using their relatives' records (3848 animals used for EBV calculation) and means that contain information from both the individual and also from relatives of the individuals. Also, G/A (n=18) genotype had a significant difference with G/G (n=64) genotype and the regression coefficient of A allele and A/A genotype shows an additive effect.

Unfortunately, at this point, we are not able to genotype more animals to increase A/A group size. Regard to this limitation, we tried to report and interpret the related results with caution (lines 261-266). We also added a paragraph to the end of dissection section to provide a more in-depth discussion of the caution associated with making inferences (Lines 417-426).

Minor points

1- It would be good if author gives explanation of their abbreviations in the manuscript for example; EBV, SNP etc...

AU: Corrected as suggested.

Reference

Han, Y., Peñagaricano, F., 2016. Unravelling the genomic architecture of bull fertility in Holstein cattle. BMC Genet. 17, 143.

Xia, Q., Li, Q., Gan, S., Guo, X., Zhang, X., Zhang, J., Chu, M., 2020. Exploring the roles of fecundity-related long non-coding RNAs and mRNAs in the adrenal glands of small-tailed Han Sheep. BMC Genet. 21, 1–11.

---

## [Decision Letter · Decision Letter 1]

9 Dec 2020

Genome-wide association study and pathway analysis identify NTRK2 as a novel candidate gene for litter size in sheep

PONE-D-20-25223R1

Dear Dr. Esmaeilifard,

We’re pleased to inform you that your manuscript has been judged scientifically suitable for publication and will be formally accepted for publication once it meets all outstanding technical requirements.

Kind regards,

Academic Editor

PLOS ONE

Additional Editor Comments (optional):

Reviewers' comments:

Reviewer's Responses to Questions

**Comments to the Author**

1. If the authors have adequately addressed your comments raised in a previous round of review and you feel that this manuscript is now acceptable for publication, you may indicate that here to bypass the “Comments to the Author” section, enter your conflict of interest statement in the “Confidential to Editor” section, and submit your "Accept" recommendation.

Reviewer #2: All comments have been addressed

Reviewer #3: All comments have been addressed

2. Is the manuscript technically sound, and do the data support the conclusions?

Reviewer #2: Yes

Reviewer #3: Yes

3. Has the statistical analysis been performed appropriately and rigorously? 

Reviewer #2: Yes

Reviewer #3: N/A

4. Have the authors made all data underlying the findings in their manuscript fully available?

Reviewer #2: Yes

Reviewer #3: Yes

5. Is the manuscript presented in an intelligible fashion and written in standard English?

Reviewer #2: Yes

Reviewer #3: Yes

6. Review Comments to the Author

Reviewer #2: (No Response)

Reviewer #3: (No Response)

7. PLOS authors have the option to publish the peer review history of their article (what does this mean?). If published, this will include your full peer review and any attached files.

Reviewer #2: No

Reviewer #3: No

---

## [Editor Report · Acceptance letter]

11 Jan 2021

PONE-D-20-25223R1 

Genome-wide association study and pathway analysis identify NTRK2 as a novel candidate gene for litter size in sheep 

Dear Dr. Esmaeili-Fard:

I'm pleased to inform you that your manuscript has been deemed suitable for publication in PLOS ONE. Congratulations! Your manuscript is now with our production department. 

Kind regards, 

on behalf of

Dr. Rajakumar Anbazhagan 

Academic Editor

PLOS ONE